# Efficacy and Safety of Trans-Arterial Yttrium-90 Radioembolization in Patients with Unresectable Liver-Dominant Metastatic or Primary Hepatic Soft Tissue Sarcomas

**DOI:** 10.3390/cancers14020324

**Published:** 2022-01-10

**Authors:** Stefano Testa, Nam Q. Bui, David S. Wang, John D. Louie, Daniel Y. Sze, Kristen N. Ganjoo

**Affiliations:** 1Department of Medicine, Stanford University, Stanford, CA 94304, USA; 2Division of Oncology, Department of Medicine, Stanford University, Stanford, CA 94304, USA; nambui@stanford.edu; 3Division of Interventional Radiology, Department of Radiology, Stanford University, Stanford, CA 94305, USA; davidwang@stanford.edu (D.S.W.); jdlouie@stanford.edu (J.D.L.); dansze@stanford.edu (D.Y.S.)

**Keywords:** trans-arterial radioembolization, soft-tissue sarcomas, Yttrium-90, liver

## Abstract

**Simple Summary:**

Sarcomas of the liver are a rare and aggressive group of malignancies for which surgery is the preferred treatment modality even though most patients are not surgical candidates and receive chemotherapy with poor outcomes. In these cases, trans-arterial liver-directed therapies are emerging as a new treatment option. Among these, radioembolization is a promising but understudied treatment option. In radioembolization, microbeads conjugated to a radioactive drug are injected into the blood vessels, nourishing the cancers and leading to cell death and tumor shrinkage. In this study, we retrospectively analyzed 35 patients with liver sarcomas receiving radioembolization at our institution. We found that those with disease control in the liver 6 months after the procedure had longer overall survival as well as patients with a liver progression-free interval post-procedure equal to or greater than 9 months. Patients with good performance status and normal liver function at baseline also had longer survival. The most common adverse reactions were nausea, fatigue, abdominal pain, and mild reversible abnormalities in liver function tests. Overall, our results suggest that radioembolization might be a safe and effective treatment option for patients with unresectable liver sarcomas.

**Abstract:**

Patients with liver-dominant metastatic or primary hepatic soft tissue sarcomas (STS) have poor prognosis. Surgery can prolong survival, but most patients are not surgical candidates, and treatment response is limited with systemic chemotherapy. Liver-directed therapies have been increasingly employed in this setting, and Yttrium-90 trans-arterial radioembolization (TARE) is an understudied yet promising treatment option. This is a retrospective analysis of 35 patients with metastatic or primary hepatic STS who underwent TARE at a single institution between 2006 and 2020. The primary outcomes that were measured were overall survival (OS), liver progression-free survival (LPFS), and radiologic tumor response. Clinical and biochemical toxicities were assessed 3 months after the procedure. Median OS was 20 months (95% CI: 13.9–26.1 months), while median LPFS was 9 months (95% CI: 6.2–11.8 months). The objective response rate was 56.7%, and the disease control rate was 80.0% by mRECIST at 3 months. The following correlated with better OS post-TARE: liver disease control (DC) at 6 months (median OS: 40 vs. 17 months, *p* = 0.007); LPFS ≥ 9 months (median OS: 50 vs. 8 months, *p* < 0.0001); ECOG status 0–1 vs. 2 (median OS: 22 vs. 6 months, *p* = 0.042); CTP class A vs. B (median OS: 22 vs. 6 months, *p* = 0.018); and TACE post-progression (median OS: 99 vs. 16 months, *p* = 0.003). The absence of metastases at diagnosis was correlated with higher median LPFS (7 vs. 1 months, *p* = 0.036). Two grade 4 (5.7%) and ten grade 3 (28.6%) laboratory toxicities were identified at 3 months. There was one case of radioembolization-induced liver disease and two cases of radiation-induced peptic ulcer disease. We concluded that TARE could be an effective and safe treatment option for patients with metastatic or primary hepatic STS with good tumor response rates, low incidence of severe toxicity, and longer survival in patients with liver disease control post-TARE.

## 1. Introduction

Soft tissue sarcomas (STS) represent an uncommon and heterogenous group of malignancies of mesenchymal origin [1,2]. The most common primary sites for STS are the extremities (60%) followed by the trunk (18%), the retroperitoneum (13%), and the head and neck (9%), with primary hepatic STS accounting for <1% of all liver cancers [3]. STS most commonly spreads to the lungs (80%), with the rate of spread to the liver varying by the site of the primary tumor and approaching up to 60% in patients with visceral sarcomas [4,5,6,7].

Patients with primary and metastatic hepatic STS have a particularly poor prognosis, with a 2-year overall survival (OS) rate of 22% for the former and 21.7% for the latter [8,9,10]. Surgical resection can improve survival in both metastatic and primary liver sarcomas [11,12,13,14]. However, many patients have unresectable disease or are not surgical candidates due to poor performance status or comorbidities. For these patients, systemic chemotherapy is employed treatment although response rates are low [15,16,17]. Thus, localized transcatheter liver-directed therapies (LDTs), including trans-arterial chemoembolization (TACE), bland embolization (TABE), and Yttrium-90 (Y90) radioembolization (TARE), offer a valuable treatment alternative for these patients. Compared to other LDTs, TARE is associated with a lower incidence of side effects, lower rates of hospitalization, and the opportunity to treat patients with extensive liver tumor burden in fewer treatment sessions [18].

Here, we review what, to our knowledge, is the largest single-institution series of patients with unresectable primary or metastatic hepatic STS treated with Y90 TARE. The goal of this study is to explore the efficacy and safety of TARE in patients with hepatic STS.

## 2. Materials and Methods

### 2.1. Patient Selection

Patients with age ≥ 18 years and either primary hepatic sarcoma or liver-dominant metastatic STS who received Y90 TARE at a single quaternary care institution between 2006 and 2020 were retrospectively analyzed. All sarcoma subtypes with or without extrahepatic disease at the time of TARE were included in the study. History of systemic chemotherapy, liver surgery, or external beam radiotherapy prior to or after TARE did not exclude patients from this study. This is because, in current clinical practice, TARE is not commonly used as the first line treatment for patients with hepatic STS, and many of these patients receive other cancer-directed therapies after TARE either due to disease progression or to achieve control of extra-hepatic disease. This study received Institutional Review Board approval, and the need for informed consent was waived. Of the 35 patients included in this study, 15 were also part of a previous multi-institutional study on patients with hepatic STS undergoing TARE [19].

### 2.2. TARE Procedure

All TARE procedures were performed by interventional radiologists with ≥10 years of experience (DYS, JDL, DSW). All patients underwent pre-procedural preparatory angiography and intra-arterial Tc99-macroaggregated albumin (Tc99-MAA) administration to simulate the predicted distribution of Y90 microspheres during TARE and to calculate the hepatopulmonary shunt fraction (HPSF). The criteria for receiving TARE were a pulmonary shunt, resulting in a less than expected dose of 30 Gy to the lungs, ECOG performance status ≤ 2, INR ≤ 1.5, serum albumin ≥ 2 g/dL, and total serum bilirubin ≤ 2 mg/dl.

### 2.3. Evaluation of Response

Radiologic tumor response was assessed on follow-up multiphasic MRI, CT, or PET/CT obtained at 3-, 6- and 9-months post-TARE. Tumor responses were classified as complete response (CR), partial response (PR), stable disease (SD), or disease progression (PD) using both the RECIST (version 1.1) and the modified RECIST (mRECIST) criteria applied only to the liver [20,21,22]. The disease control rate (DC) was calculated at each time point as the proportion of patients achieving either CR, PR, or SD among all patients treated and based on which imaging was available. The objective response rate (ORR) was defined as the proportion of patients achieving either CR or PR among all patients treated and based on which imaging was available.

### 2.4. Evaluation of Toxicity

Clinical and biochemical toxicities were assessed at baseline and 3 months after TARE. All adverse reactions were categorized using the Common Terminology Criteria for Adverse Events (CTCAE) version 5.0 [23]. The following laboratory tests were available at baseline and 3 months post-TARE for all the patients in the study: aspartate aminotransferase (AST), alanine aminotransferase (ALT), total bilirubin, alkaline phosphatase, INR, PT, PTT, serum sodium, serum creatinine, and serum albumin. The model for end-stage liver disease (MELD) score was calculated at baseline and 3 months post-procedure. The Albumin–bilirubin score (ALBI) and Child–Turcotte–Pugh class (CTP) were measured at baseline. All patients were evaluated with a clinical visit with Interventional Radiology at 1 and 3 months after the procedure. Potential clinical adverse events were screened for at these visits.

### 2.5. Statistical Analysis

Statistical analyses were performed using SPSS version 27.0 (IBM). The endpoints that were measured were overall survival (OS), liver progression-free survival (LPFS), radiographic tumor response, and adverse events. OS was calculated from the date of TARE to the date of death or was censored at the date of the last follow-up. LPFS was calculated from the date of TARE to the date of first detected liver disease progression or was censored at the time of death or at the last follow-up if the patient had never experienced liver disease progression beforehand. The Kaplan–Meier method was used to calculate the OS and LPFS. The log-rank test was employed to detect differences between survival curves. The univariate Cox proportional hazard model was used to measure the association of different variables with OS and LPFS. The variables that were the most strongly associated with the OS in the univariate analysis were included in the multivariate Cox proportional hazard analysis. The paired Student’s t-test was used to assess the difference between the mean MELD scores at baseline and at 3 months post-TARE. The Chi-square test was used to calculate differences in tumor response when assessed with the RECIST or mRECIST criteria.

## 3. Results

### 3.1. Patient Baseline Characteristics

Baseline patient characteristics are described in Table 1. A total of 35 patients were included, with a median age of 58 years (range 24–83 years), with of the patients being 21 females (60%). Most patients had a pre-TARE ECOG performance status of 0–1 (*n* = 29, 82.8%), ALBI grade of 2–3 (*n* = 22, 62.9%), CTP class A (*n* = 22, 77.1%), and median MELD score of 8 (range of 6–19). The most common sarcoma subtype was leiomyosarcoma (*n* = 20, 57.1%, LMS). The retroperitoneum was the most common primary site (*n* = 9, 25.7%) followed by the liver (*n* = 7, 20%). Among primary liver sarcomas, the most common subtypes were epithelioid hemangioendothelioma (*n* = 2, 5.7%) and hepatic LMS (*n* = 2, 5.7%). Most patients had tumors involving both lobes of the liver at the time of TARE (*n* = 29, 82.9%) with large hepatic tumor burden and conglomerated masses or infiltrative disease that precluded the accurate measurement of the tumor volumes. The mean total liver volume at the time of TARE was 2.4 L (SD ± 1.8 L), where the average volume for non-cirrhotic liver in adults is 1.5 L ± 0.38 L [24]. Additionally, more than half of the patients had extrahepatic metastases at the time of TARE, both among those with primary hepatic STS (*n* = 4, 57.1%) and those with metastatic STS (*n* = 18, 64.2%), even though the liver was the most active disease progression site.

As far as treatments before TARE, 27 patients (77.1%) received systemic chemotherapy, 9 (25.7%) received liver surgery, and 3 (8.6%) received other LDTs such as TACE, percutaneous ethanol ablation, microwave ablation, and radiofrequency ablation. No patients received external beam radiation to the liver, while 16 patients (45.7%) received radiotherapy directed to the primary tumor.

### 3.2. Treatment

Most of the patients in the study (*n* = 32, 91.4%) received prophylactic coil embolization of non-target arteries to prevent the delivery of Y90 microspheres to the stomach and duodenum. The Mean Hepato-Pulmonary Shunt Fraction (HPSF) was 9.3% (SD ± 8.2%), with a median of 6.9% (range 1.3–27.7%). Thirty-one patients (88.6%) were treated with Y90 resin microspheres (SIR-Spheres, Sirtex, Woburn, MA, USA) and 4 (11.4%) with Y90 glass microspheres (TheraSphere, Boston Scientific, Marlborough, MA, USA) due to either portal vein occlusion (*n* = 2, 5.7%) or due to the poor flow of the common hepatic artery (*n* = 2, 5.7%). The median activity administered was 1.87 GBq (range 0.94–5.25 GBq). The median dose to the liver, calculated by the Medical Internal Radiation Dose (MIRD) method, was 54.9 Gy (range 17.8–335.8 Gy) [25]. Four patients received a higher dose delivered to <2 liver segments (radiation segmentectomy), with a median dose to the liver of 121.8 Gy (range 105.1–335.8 Gy). The median dose to the lung in a single treatment was 8.5 Gy (range 0.6–28.5 Gy). Most patients underwent bilobar TARE (*n* = 25, 71.4%), with only 10 patients (28.6%) receiving unilobar or segmental TARE. Once they experienced liver disease progression after TARE, five patients (14.3%) received a repeat TARE, while six (17.1%) received TACE using either doxorubicin-eluting microspheres (*n* = 4, 66.7%), doxorubicin-lipiodol emulsion (*n* = 1, 16.7%), or irinotecan-eluting microspheres (*n* = 1, 16.7%).

### 3.3. Radiologic Tumor Response and Follow-Up

The median follow-up post-TARE was 16 months (range 2–132 months). Follow-up imaging was available for 30 patients at 3 months, for 25 patients at 6 months, and for 15 patients at 9 months after TARE (Figure 1). The most common imaging modality that was employed to assess liver disease response at 3 months was triphasic liver CT (*n* = 15), followed by PET/CT (*n* = 10) and liver MRI (*n* = 5). When response was assessed by the mRECIST criteria at 3 months post-TARE, we observed a DC rate of 80.0% and an ORR of 56.7% (Table 2). Additionally, at each time point, there was no difference in the DC rate and in the proportion of patients with PD when assessed with the RECIST compared to with the mRECIST criteria, while the ORR was higher when it was measured by the mRECIST compared to with the RECIST criteria at 3- and 6 months, but not at 9 months (*p* = 0.0004, *p* = 0.02, *p* = 0.09, respectively). To assess if other cancer-directed treatments before or after TARE had an impact on radiologic tumor response, we compared the tumor response between patients who received and those who did not receive the following therapies: chemotherapy pre-TARE; different LDTs pre-TARE; liver surgery pre-TARE; external beam radiotherapy pre-TARE; and chemotherapy post-TARE (Appendix A). We found that receiving any of the above treatments did not result in differences in the DC rate, ORR, or proportion of patients with PD at 3-, 6- and 9-months post-TARE, regardless of if they were measured by RECIST or mRECIST criteria. The sole exception was receiving liver surgery pre-TARE, which only correlated with a lower DC rate when the tumor response was measured by the mRECIST criteria but not by the RECIST criteria.

### 3.4. Overall Survival and Liver Progression-Free Survival

The median OS after TARE was 20 months (95% CI: 13.9–26.1 months), with an OS rate of 62%, 34%, 17%, and 4% at 1, 3, 5, and 10 years, respectively (Figure 2A). The median LPFS was 9 months (95% CI: 6.2–11.8 months, Figure 2B), with a LPFS rate of 69%, 39%, 25%, and 8% at 6 months and at 1, 2, and 5 years, respectively. The impact of the different variables on the OS and LPFS is reported through both Log-rank and univariate Cox proportional hazards analyses (Table 3 and Table 4).

When response was measured through the RECIST criteria, patients with liver DC at 3 and 9 months had a similar median OS compared to those with liver PD. Instead, at 6 months, patients with liver DC had a higher median OS than patients with liver PD (40 months, 95% CI: 17.6–62.4 vs. 17 months, 95% CI: 8.1–25.9, *p* = 0.007, Figure 2C) as well as a reduced risk of death (HR: 0.25, 95% CI: 0.08–0.75. *p* = 0.013). Similar results were observed when the response at 3, 6, and 9 months was assessed through the mRECIST criteria (Appendix A). The patients who experienced LPFS ≥ 9 months had a higher median OS compared to those with LPFS < 9 months (50 months, 95%CI: 30.8–69.1 vs. 8 months, 95%CI: 5.0–10.9, *p* < 0.0001, Figure 2D) together with a lower risk of death post-TARE (HR: 0.14, 95%CI: 0.04–0.4, *p* < 0.0001).

The presence versus absence of extrahepatic disease at the time of TARE did not result in a different median LPFS, median OS, risk of death (HR: 1.7, 95%CI: 0.8–3.8, *p* = 0.197), or risk of liver disease progression (HR: 1.1, 95%CI: 0.5–2.7, *p* = 0.754). Patients with localized disease at diagnosis had a longer median LPFS than those with metastases at diagnosis (19 months, 95%CI: 4.1–33.9 vs. 7 months, 95%CI: 3.4–10.6, *p* = 0.036) as well as an increased risk of liver disease progression after TARE (HR: 2.5, 95%CI: 1.1–6.2, *p* = 0.048), with no differences being observed in either median OS or risk of death post-TARE between the two groups.

Patients treated with resin Y90 microspheres and those treated with glass Y90 microspheres had a similar median LPFS, median OS, risk of liver disease progression and risk of death. In addition, patients who had bilobar TARE had a similar OS, LPFS, risk of death (HR: 0.57, 95%CI: 0.25–1.3, *p* = 0.193), and risk of liver disease progression (HR: 0.9, 95%CI: 0.3–2.2, *p* = 0.852) compared to those who received unilobar or segmental TARE. Of note, the patients who received segmental TARE and those who received TARE to a single hepatic lobe had a similar median LPFS, median OS, risk of death, and risk of liver disease progression post-TARE. Additionally, the median OS and LPFS, risk of death, and risk of liver disease progression were comparable between patients with bilobar liver disease and those with unilobar disease as well as between patients with a liver volume greater or lower than 1.5 L at baseline.

Patients with a baseline ECOG performance status of 2 had a shorter median OS than those with an ECOG status of 0–1 (6 months, 95% CI: 0.0–12.3 vs. 22 months, 95% CI: 14.2–29.8, *p* = 0.042) together with a higher risk of death (HR: 2.7, 95%CI: 0.9–7.2, *p* = 0.054) but with a similar median LPFS and risk of liver disease progression. Patients with baseline CTP class B had a shorter median OS compared to those with baseline CTP class A (6 months, 95%CI: 4.7–7.3 vs. 22 months, 95%CI: 13.1–30.9, *p* = 0.018) as well as an increased risk of death post-TARE (HR: 2.6, 95%CI: 1.1–6.2, *p* = 0.025) but had a similar LPFS and risk of liver disease progression. Additionally, there was no difference in the OS, LPFS, risk of death, or risk of liver disease progression between patients with a baseline MELD > 9 and those with a baseline MELD ≤ 9 as well as between patients with a baseline ALBI score of grades 2–3 and those with a baseline ALBI score of grade 1.

Finally, patients with LMS and those with a non-LMS sarcoma had a similar median OS, median LPFS, risk of death (HR: 0.71, 95%CI: 0.3–1.6, *p* = 0.413), and risk of liver disease progression (HR: 0.6, 95%CI: 0.3–1.6, *p* = 0.337).

### 3.5. The Impact of Pre-TARE and Post-TARE Therapy on OS and LPFS

Patients that received TACE post-TARE after experiencing disease progression in the liver had a longer median OS compared to those who did not receive TACE (99 months, 95%CI: 90.9–107.0 vs. 16 months, 95%CI: 4.9–27.0, *p* = 0.003) as well as a lower risk of death (HR: 0.14, 95%CI: 0.03–0.6, *p* = 0.009). The median interval of time between TARE and TACE was 7 months (range 4–52 months). Instead, patients who received a second TARE after liver disease progression (*n* = 5, 14.3%) had a similar median OS compared to those who did not (33 months, 95%CI: 3.7–30.3 vs. 17 months, 95%CI: 4.9–27.0, *p* = 0.517) together with a similar risk of death (HR: 0.73, 95%CI:0.3–1.9, *p* = 0.524). The median interval of time between the first and second TARE was 10 months (range 5–76 months). In four patients, the repeat TARE was administered to lesions recurring in areas that were treated during the first radioembolization, while only one patient received the second TARE to a new lesion occurring in a liver territory that had not been previously treated.

Patients who received systemic chemotherapy between TARE and the first liver disease progression (interim chemotherapy) had a similar median LPFS compared to those who did not receive interim chemotherapy (8 months, 95%CI: 6.5–9.5 vs. 10 months, 95%CI: 0.0–22.5, *p* = 0.463) as well as similar risk of liver disease progression (HR: 1.4, 95%CI: 0.6–3.3, *p* = 0.477). Patients who received chemotherapy post-TARE, either before or after the first liver disease progression, had a similar median OS and risk of death post-procedure compared to patients who did not receive chemotherapy after TARE. Additionally, receiving chemotherapy, radiotherapy, liver surgery, or other LDTs before TARE did not affect the median OS, median LPFS, risk of liver disease progression, or risk of death after TARE.

### 3.6. Multivariate Analysis

Only the variables that correlated with better OS in both the Kaplan–Meier and the univariate Cox proportional hazards analyses were included in the multivariate Cox proportional hazards analysis of the OS. In this case, both receiving TACE after TARE (HR: 0.17, 95%CI: 0.04–0.78, *p* = 0.023) and a liver PFS ≥ 9 months (HR: 0.16, 95%CI: 0.05–0.50, *p* = 0.002) maintained their association with a reduced risk of death post-TARE, while the opposite was true for a baseline CTP class B (HR: 1.5, 95%CI: 0.63–3.65, *p* = 0.344) (Table 5). The RECIST response status at 6 months was not included in the multivariate analysis since the number of uncensored events at 6 months post-TARE was 19, which would have resulted in approximately 6 events per variable and consequent issues in data analysis and interpretation.

### 3.7. Toxicity

The most common symptom after TARE was fatigue (*n* = 17) followed by nausea (*n* = 13) and right upper quadrant abdominal pain (*n* = 6) (Table 6). One patient experienced radioembolization-induced liver disease (REILD) that presented as an elevation in alkaline phosphatase, gamma-glutamyl transferase, and total bilirubin with worsening ascites 7 weeks after TARE [26]. This patient was started on prednisone and ursodiol, which resulted in an improvement in liver function and the resolution of laboratory abnormalities over the course of a month.

One patient developed interstitial pneumonia 4 months after radioembolization with symptoms that started soon after chemoembolization, which suggested a radiation-recall pneumonitis rather than a radiation-induced pneumonitis. This patient required a short inpatient stay for supplemental oxygen administration with improvement after a course of high-dose corticosteroids.

Two patients developed gastroduodenal peptic ulcer disease (PUD) that manifested as epigastric pain, nausea, and dyspepsia without gastrointestinal bleeding. Both patients received prophylactic coil embolization of hepato-enteric collaterals that had been identified during the preparatory angiography and had complete recovery after a two-month course of oral proton pump inhibitors [27].

As far as laboratory abnormalities, at 3 months post-TARE, we observed new grade 3 (*n* = 2, 5.7%) and grade 1 (*n*= 16, 45.7%) liver transaminases elevations, and of the latter two occurred in patients with concomitant liver disease progression. We also observed new grade 4 (*n* = 2, 5.7%), grade 3 (*n* = 2, 5.7%), and grade 2 (*n* = 12, 34.3%) hyperbilirubinemia, of which one, none, and two, respectively occurred in patients with concomitant liver disease progression. There was also new grade 2 alkaline phosphatase elevation (*n* = 3, 8.6%), of which one case occurred in a patient with concomitant liver disease progression, and grade 1 alkaline phosphatase elevation (*n* = 2, 5.7%). Additionally, we noted new grade 3 (*n* = 4, 11.4%) and grade 2 (*n* = 5, 14.3%) hypoalbuminemia, and of the latter two occurred in patients with concomitant liver disease progression. We noted grade 3 (*n* = 1, 2.8%) and grade 2 (*n* = 3, 8.6%) INR elevation as well. The above laboratory abnormalities resolved between 1- and 9-months post-TARE. Finally, there was a statistically significant increase in the mean MELD score 3 months post TARE compared to at baseline (11.7 vs. 9.2, MD:2.5, 95%CI: 0.6–4.4, *p* = 0.001).

## 4. Discussion

Transcatheter liver-directed therapies (LDTs) have been proven to increase survival in patients with hepatocellular carcinoma, cholangiocarcinoma, as well as colorectal cancer (CRC) and neuroendocrine tumors with liver-dominant metastases [28,29,30,31]. However, there are very few studies describing the efficacy and safety of LDTs in patients with primary or metastatic hepatic sarcomas. Most of these studies focus on TACE or TABE, reporting modest rates of tumor response and improved survival [32,33,34,35]. The only data available for TARE in hepatic sarcomas come from one multi-institutional retrospective study and case reports [19,36,37]. However, TARE is particularly appealing due to the lower incidence of post-embolization syndrome and hospital stays of only 2–3 h compared to TACE and other LDTs that often require hospitalization for 1–3 days [38]. Here, we describe what, to our knowledge, is the largest single-institution series of patients with primary or metastatic hepatic STS receiving Y90 TARE.

We observed a median OS of 20 months and a LPFS of 9 months, which is similar to prior studies of LDTs in hepatic sarcomas. Chapiro et al. showed a median OS from TACE of 21.2 months (95% CI, 13.4–28.9 months), with a median LPFS of 6.3 months (95% CI, 4.4–8.2 months) in patients with liver-dominant metastatic STS [32]. Another study with TACE in patients with secondary hepatic STS showed a median OS of 13 months, with OS of 67% and 40% at 1- and 3 years post-procedure, respectively [33]. Additional studies with TABE, TACE, TARE, or microwave ablation showed median OS ranging from 9 to 26.7 months and median LPFS of 9 to 14.2 months [34,35]. A study from Miller et al. on TARE in patients with primary or metastatic liver sarcomas showed a median OS of 30 months (95% CI: 12–43 months), with 1-year and 3-year OS rates of 83% and 37%, respectively [19]. The longer OS observed in that study compared to in ours may be due to the heterogeneity in the patient population across different institutions.

We also found that patients with DC at 6 months, either by the RECIST or mRECIST criteria, had longer survival compared to patients with PD, similar to prior retrospective studies where a favorable radiologic response to LDTs in patients with liver-dominant metastatic STS and HCC was associated with better survival [19,39,40]. We also found that patients who achieved LPFS ≥ 9 months had longer survival compared to patients with LPFS < 9 months, indicating that LPFS after TARE might be a good surrogate for OS in patients with hepatic STS even though this was not the case for patients with CRC that was metastatic to the liver [41,42].

Additionally, in this study, patients with extrahepatic disease at the time of TARE had similar survival to those with disease limited to the liver, which is in contrast to what was observed in metastatic CRC where the presence of extrahepatic disease significantly the reduced survival of patients undergoing TARE [41,42]. This suggests that TARE might be clinically useful in patients with hepatic STS, even among those with extrahepatic dissemination of disease.

We also found that patients with a baseline ECOG status of 0–1 had a longer survival post-TARE, which is similar to prior studies [43,44,45]. Additionally, patients with CTP class B had a worse median OS and increased risk of death than patients with CTP class A, reinforcing the notion that TARE should be considered in patients with good baseline hepatic function.

We found that patients who received TACE at liver disease progression post-TARE had longer survival compared to those who did not. Instead, receiving a second TARE at liver disease progression was not associated with improved survival. Overall, these results suggest that TACE may be a better option compared to a second TARE following liver disease progression, or that TACE should be selected for patients with more limited or indolent progression. Definitive recommendations on which second-line LDT should be employed after TARE cannot be made due to the small sample size and the possible selection bias where only patients with a more indolent course lived long enough to be able to undergo TACE or to repeat TARE.

Prior chemotherapy, external beam radiation, liver surgery, or a different LDT before TARE did not affect survival after TARE, suggesting that prior treatments might not be a factor to weigh heavily when deciding which patients with hepatic STS should undergo TARE.

In this study, TARE had a favorable toxicity profile compared to previous reports [46,47]. The most common symptoms were fatigue, nausea, and right upper quadrant abdominal pain. The majority of laboratory adverse events were mild and reversible liver function test abnormalities, which was also similar to prior retrospective series [19,34]. We had one case of REILD in a patient with normal liver function at baseline who had TARE of the entire liver and systemic chemotherapy 3 weeks after TARE [48]. There were two cases of grade 2 radiation-associated PUD, and these occurred in patients who had been treated earlier on in our experience and received gastroduodenal artery coil embolization, which we currently do not routinely perform on all patients undergoing TARE [47].

Lastly, this study has several limitations, including its small sample size and multiple sarcoma subtypes, which is a consequence of the rarity of these malignancies, with some subtypes showing a more indolent course such as epithelioid hemangioendothelioma and hemangiopericytoma. In addition, patient selection in our study was not standardized but was instead based on the clinical judgment of both the referring oncologist and the interventional radiologist. Lastly, since this is a single-institution study, our results might be difficult to translate to the general population due to differences in patient characteristics and the variability in treatment practices at other institutions.

## 5. Conclusions

In summary, our study indicates that TARE could be an effective and safe treatment option for patients with unresectable liver-dominant primary or metastatic hepatic STS. Our results suggest that the patients who could the most benefit from the procedure are those with a baseline ECOG status of 0–1 and CTP class A. Additionally, the presence of extrahepatic disease at the time of TARE did not correlate with worse survival, suggesting that TARE might even be beneficial for patients with hepatic STS and concomitant extrahepatic disease. Finally, our study shows that a good radiologic tumor response post-TARE positively correlates with survival and the fact that LPFS could be used as a surrogate marker for OS in patients with hepatic STS.

However, due to the small size of our patient cohort and the single-institution nature of this study, we cannot make definitive conclusions regarding the safety and efficacy of TARE in patients with hepatic STS. In contrast, our results could lay the basis for further, larger multi-institutional randomized studies that will be better able to delineate the therapeutic role of TARE in patients with unresectable hepatic STS.

## Figures and Tables

**Figure 1 cancers-14-00324-f001:**
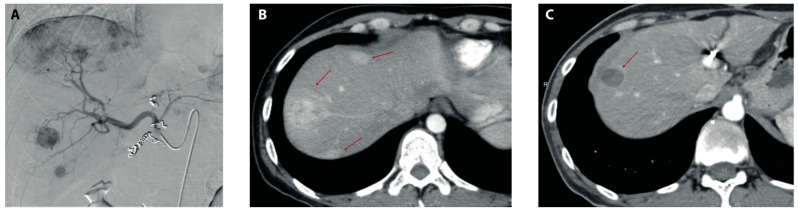
Imaging before and after TARE. (**A**) Preparatory angiography pre-TARE showing several contrast-enhancing lesions in the right hepatic lobe after the administration of iodinated contrast in the common hepatic artery. (**B**) Arterial phase of triphasic liver CT scan showing three arterially enhanced lesions (red arrows) in the right hepatic lobe 2 months before TARE. (**C**) Arterial phase of triphasic CT scan 9 months post-TARE showing significant reductions in both size and the contrast-enhancement of dominant lesions in the right hepatic lobe (red arrow) with the disappearance of minor lesions previously observed in (**B**).

**Figure 2 cancers-14-00324-f002:**
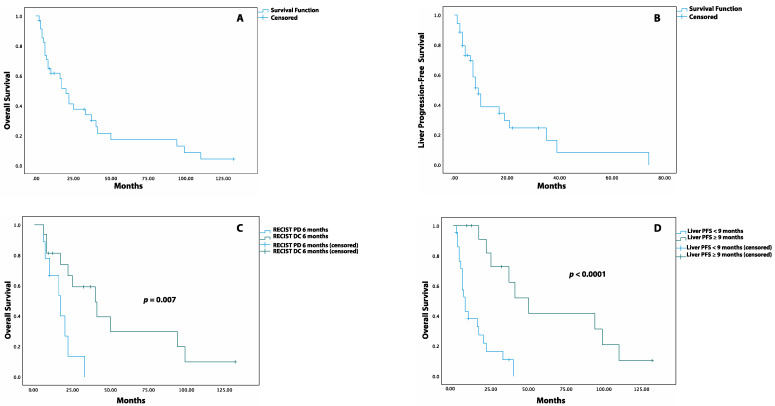
Kaplan–Meier Survival Curves for OS and LPFS post-TARE. (**A**) Overall survival after TARE. (**B**) Liver progression-free survival after TARE. (**C**) Overall survival after TARE in patients with DC (disease control) versus PD (progression of disease) at 6 months when assessed by RECIST criteria. (**D**) Overall survival after TARE in patients with liver PFS equal to or greater than 9 months versus patients with liver PFS lower than 9 months.

**Table 1 cancers-14-00324-t001:** Patient characteristics and demographics.

Patient Characteristic	Number (%)
Age at TARE	
Median Range	58 (24–83)
Sex	
Male Female	14 (40%) 21 (60%)
Baseline ECOG Performance Status	
0 1 2	16 (45.7%) 13 (37.2%) 6 (17.1%)
Primary Site	
Retroperitoneum Liver Uterus Extremities Lung Stomach Other	9 (25.7%) 7 (20%) 4 (11.4%) 3 (8.6%) 2 (5.7%) 2 (5.7%) 8 (22.9%)
Histotype	
Leiomyosarcoma Spindle Cell Sarcoma Hemangiopericytoma Hepatic Epithelioid Hemangioendothelioma Gastro-Intestinal Stromal Tumor Angiosarcoma Rhabdomyosarcoma Liposarcoma	20 (57.1%) 4 (11.4%) 3 (8.6%) 2 (5.7%) 2 (5.7%) 2 (5.7%) 1 (2.9%) 1 (2.9%)
Baseline MELD	
Median Range	8 (6–19)
Baseline Child-Turcotte-Pugh Class	
Class A Class B	27 (77.1%) 8 (22.9%)
Baseline ALBI grade	
Grade 1 Grade 2–3	13 (37.1%) 22 (62.9%)
Primary Metastases	
Yes No	16 (45.7%) 19 (54.3%)
Extrahepatic Disease	
Yes No	22 (62.9%) 13 (37.1%)
Chemotherapy Pre-TARE	
Yes No	27 (77.1%) 8 (22.9%)
Liver Surgery Pre-TARE	
Yes No	9 (25.7%) 26 (74.3%)
Radiotherapy Pre-TARE	
Yes No	16 (45.7%) 19 (54.3%)
Liver Involvement	
Bilobar Single lobe	29 (82.8%) 6 (17.1%)
Baseline Liver Volume	
>1.5 L ≤1.5 L	25 (71.4%) 10 (28.6%)

TARE, trans-arterial radioembolization; ECOG, Eastern Cooperative Oncology Groups; ALBI, albumin–bilirubin; MELD, Model for End-Stage Liver Disease.

**Table 2 cancers-14-00324-t002:** Tumor response post-TARE.

Radiologic Tumor Response	RECIST	mRECIST	*p*-Value (χ^2^)
3 months post-TARE			
DC (CR + PR + SD)	73.3% (22/30)	80.0% (24/30)	0.54
ORR (CR + PR)	13.3% (4/30)	56.7% (17/30)	0.0004
PD	26.7% (8/30)	20.0% (6/30)	
6 months post-TARE			
DC (CR + PR + SD)	64.0% (16/25)	68.0% (17/25)	0.76
ORR (CR + PR)	20.0% (5/25)	52.0% (13/25)	0.02
PD	36.0% (9/25)	32.0% (8/25)	
9 months post-TARE			
DC (CR + PR + SD)	40.0% (6/15)	46.7% (7/15)	0.46
ORR (CR + PR)	13.3% (2/15)	40.0% (6/15)	0.09
PD	60.0% (9/15)	53.3% (8/15)	

TARE, trans-arterial radioembolization; RECIST, Response Evaluation Criteria in Solid Tumors; mRECIST, Modified Response Evaluation Criteria in Solid Tumors; DC, disease control; CR, complete response; PR, partial response; SD, stable disease; PD, disease progression; ORR, objective response rate; (χ^2^), Chi-square test.

**Table 3 cancers-14-00324-t003:** Univariate analysis of overall survival.

Variables	Number (%)	Kaplan–Meier Median OS (Months, 95%CI)	*p*-Value (Log-Rank)	Cox Proportional Hazards Hazard Ratio (95%CI)	*p*-Value (Cox)
RECIST response 3 months					
DC PD	22 (62.8%) 8 (22.8%)	22 (10.9–33.1) 10 (0.0–21.9)	0.053	0.41 (0.16–1.1)	0.065

RECIST response 6 months					
DC PD	16 (45.7%) 9 (25.7%)	40 (17.6–62.4) 17 (8.1–25.9)	0.007	0.25 (0.08–0.75)	0.013

RECIST response 9 months					
DC PD	6 (17.1%) 9 (25.7%)	94 (0.0–242.1) 37 (26.7–47.3)	0.290	0.47 (0.1–1.9)	0.299

Liver PFS					
≥9 months <9 months	13 (37.1%) 22 (62.9%)	50 (30.8–69.1) 8 (5.0–10.9)	<0.0001	0.14 (0.04–0.4)	<0.0001

TACE post-TARE					
Yes No	6 (17.1%) 29 (82.9%)	99 (90.9–107.0) 16 (4.9–27.0)	0.003	0.14 (0.03–0.6)	0.009

Repeat TARE					
Yes No	5 (14.3%) 30 (85.7%)	33 (3.7–30.3) 17 (4.9–27.0)	0.517	0.73 (0.3–1.9)	0.524

Primary Metastases					
Yes No	16 (45.7%) 19 (54.3%)	17 (11.1–22.9) 22 (0.0–57.5)	0.173	1.8 (0.8–4.1)	0.185

Extrahepatic Metastases					
Yes No	22 (62.8%) 13 (37.2%)	17 (1.6–32.4) 37 (4.6–69.4)	0.186	1.7 (0.8–3.8)	0.197

Liver Involvement					
Bilobar Single lobe	29 (82.8%) 6 (17.1%)	22 (7.9–35.6) 16 (0.0–35.6)	0.348	0.6 (0.2–1.7)	0.359

Baseline Liver Volume					
>1.5 L ≤1.5 L	25 (71.4%) 10 (28.6%)	17 (2.0–32.0) 20 (0.0–49.3)	0.474	1.3 (0.6–3.0)	0.481

Child-Turcotte-Pugh Class					
B A	8 (22.9%) 27 (77.1%)	6 (4.7–7.3) 22 (13.1–30.9)	0.018	2.6 (1.1–6.2)	0.025

Baseline MELD					
>9 ≤9	8 (22.9%) 27 (77.1%)	33 (0.0–91.1) 17 (11.2–22.8)	0.668	1.2 (0.5–2.9)	0.673

Baseline ALBI grade					
Grade 2–3 Grade 1	22 (62.9%) 13 (37.1%)	16 (2.6–29.4) 33 (10.4–55.6)	0.308	1.5 (0.7–3.4)	0.318

Chemotherapy Post-TARE					
Yes No	22 (62.9%) 13 (37.1%)	17 (1.9–32.1) 22 (13.7–30.3)	0.773	1.1 (0.5–2.4)	0.776

Chemotherapy Pre-TARE					
Yes No	27 (77.1%) 8 (22.9%)	17 (3.1–30.9) 33 (9.6–56.4)	0.619	1.2 (0.5–3.2)	0.624

Liver Surgery Pre-TARE					
Yes No	9 (25.7%) 26 (74.3%)	20 (11.2–28.8) 22 (3.3–40.7)	0.851	0.92 (0.41–2.1)	0.853

Radiotherapy Pre-TARE					
Yes No	16 (45.7%) 19 (54.3%)	8 (6.1–9.9) 25 (3.5–46.5)	0.244	1.5 (0.73–3.3)	0.254

Liver-directed therapy Pre-TARE					
Yes No	3 (8.6%) 32 (91.4%)	37 (3.4–70.6) 20 (8.7–31.3)	0.598	0.68 (0.16–2.9)	0.606

Type of Y90 Microspheres					
Resin (SIR) Glass (Theraspheres)	31 (88.6%) 4 (11.4%)	20 (13.0–26.9) 17 (0.0–46.4)	0.631	0.77 (0.3–2.3)	0.637

TARE Distribution					
Bilobar Single Lobe/Segmental	22 (62.8%) 13 (37.2%)	22 (10.9–33.1) 16 (0.5–31.6)	0.181	0.57 (0.25–1.3)	0.193

Baseline ECOG					
2 0–1	6 (17.2%) 29 (82.8%)	6 (0.0–12.3) 22 (14.2–29.8)	0.042	2.7 (0.9–7.2)	0.054

Histotype					
Leiomyosarcoma Other	20 (57.1%) 15 (42.9%)	20 (8.1–31.9) 16 (0.0–53.6)	0.405	0.71 (0.3–1.6)	0.413


ECOG, Eastern Cooperative Oncology Groups; ALBI, albumin–bilirubin; MELD, Model for End-Stage Liver Disease; PFS, progression-free survival; RECIST, Response Evaluation Criteria in Solid Tumors; DC, disease control; PD, disease progression; TARE, trans-arterial radioembolization; Y90, Yttrium-90; TACE, trans-arterial chemioembolization; 95%CI, 95% confidence interval.

**Table 4 cancers-14-00324-t004:** Univariate analysis of liver progression-free survival.

Variables	Number (%)	Kaplan–Meier Median LPFS (Months, 95%CI)	*p*-Value (Log-Rank)	Cox Proportional Hazards Hazard Ration (95%CI)	*p*-Value (Cox)
Primary Metastases					
Yes No	16 (45.7%) 19 (54.3%)	7 (3.4–10.6) 19 (4.1–33.9)	0.036	2.5 (1.1–6.2)	0.048

Extrahepatic Metastases					
Yes No	22 (62.8%) 13 (37.2%)	8 (3.5–12.5) 9 (5.8–12.2)	0.748	1.1 (0.5–2.7)	0.754

Liver Involvement					
Bilobar Single lobe	29 (82.8%) 6 (17.1%)	9 (6.4–11.6) 4 (1.9–6.1)	0.815	0.9 (0.3–2.7)	0.818

Baseline Liver Volume					
>1.5 L ≤1.5 L	25 (71.4%) 10 (28.6%)	10 (7.6–12.4) 7 (0.8–13.2)	0.838	1.1 (0.4–2.9)	0.842

Child-Turcotte-Pugh Class					
B A	8 (22.9%) 27 (77.1%)	39 (N/A) 9 (6.3–11.7)	0.892	1.1 (0.3–3.3)	0.894

Baseline MELD					
>9 ≤9	8 (22.9%) 27 (77.1%)	9 (0.0–18.8) 8 (4.9–11.1)	0.984	1.0 (0.4–2.8)	0.984

Baseline ALBI grade					
Grade 2–3 Grade 1	22 (62.9%) 13 (37.1%)	8 (4.4–11.6) 10 (6.1–13.9)	0.508	0.8 (0.3–1.8)	0.521

Interim Chemotherapy					
Yes No	14 (40%) 21 (60%)	8 (6.5–9.5) 10 (0.0–22.5)	0.463	1.4 (0.6–3.3)	0.477

Chemotherapy Pre-TARE					
Yes No	27 (77.1%) 8 (22.9%)	9 (6.1–11.9) 8 (0.0–37.8)	0.416	1.5 (0.5–4.2)	0.429

Liver Surgery Pre-TARE					
Yes No	9 (25.7%) 26 (74.3%)	10 (0.0–24.6) 9 (6.2–11.8)	0.595	1.3 (0.5–3.1)	0.606

Radiotherapy Pre-TARE					
Yes No	16 (45.7%) 19 (54.3%)	8 (4.9–11.0) 10 (4.7–15.4)	0.713	0.8 (0.4–1.9)	0.720

Liver-directed therapy Pre-TARE					
Yes No	3 (8.6%) 32 (91.4%)	8 (1.6–14.4) 10 (5.7–14.3)	0.248	2.0 (0.6–7.1)	0.272

Type of Y90 Microspheres					
Resin (SIR) Glass (Theraspheres)	31 (88.6%) 4 (11.4%)	9 (6.4–11.6) 7 (2.2–11.8)	0.911	1.1 (0.3–3.8)	0.913

TARE Distribution					
Bilobar Single Lobe/Segmental	22 (62.8%) 13 (37.2%)	10 (7.4–12.6) 8 (2.9–13.1)	0.848	0.9 (0.3–2.2)	0.852

Baseline ECOG					
2 0–1	6 (17.2%) 29 (82.8%)	8 (N/A) 10 (7.1–10.8)	0.148	2.5 (0.7–9.1)	0.173

Histotype					
Leiomyosarcoma Other	20 (57.1%) 15 (42.9%)	8 (6.4–9.6) 10 (0.0–22.6)	0.320	0.6 (0.3–1.6)	0.337


ECOG, Eastern Cooperative Oncology Groups; ALBI, albumin–bilirubin; MELD, Model for End-Stage Liver Disease; N/A, not applicable; Interim Chemotherapy, chemotherapy administered between TARE and the first detected liver disease progression; TARE, trans-arterial radioembolization; Y90, Yttrium-90; 95%CI, 95% confidence interval.

**Table 5 cancers-14-00324-t005:** Multivariate Cox proportional hazards analysis for OS from TARE.

Variables	Hazard Ratio	95% Confidence Interval	*p*-Value
Liver PFS			
≥9 months	0.16	(0.05–0.50)	0.002
<9 months			
TACE post-TARE			
Yes	0.17	(0.04–0.78)	0.023
No			
Child-Turcotte-Pugh			
B	1.5	(0.63–3.65)	0.344
A			

PFS, Progression-free survival; TACE, Trans-arterial chemo-embolization; TARE, trans-arterial radioembolization.

**Table 6 cancers-14-00324-t006:** Analysis of toxicity and adverse reactions.

Adverse Reaction	Baseline				3 Months Post-TARE			
	Grade 1	Grade 2	Grade 3	Grade 4	Grade 1	Grade 2	Grade 3	Grade 4
Laboratory Abnormalities								
AST Elevation	9				10		1	
ALT Elevation	6				6		1	
Hypoalbuminemia	9	7			7	10	4	
Elevated Alkaline Phosphatase	16	5			4	4		
Hyperbilirubinemia	5				5	12	2	2
Hyponatremia	4				5	1		
Elevated INR	3				1	3	2	
Complications								
REILD							1	
PUD						2		

AST, aspartate aminotransferase; ALT, alanine aminotransferase; INR, international normalized ratio; TARE, trans-arterial radioembolization; REILD, radioembolization-induced liver disease; PUD, peptic ulcer disease.

## Data Availability

The data and supporting findings of this study are available from the corresponding author, S.T., upon reasonable request.

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
