# Peer review of "Efficacy and Safety of Trans-Arterial Yttrium-90 Radioembolization in Patients with Unresectable Liver-Dominant Metastatic or Primary Hepatic Soft Tissue Sarcomas"

_cancers, 2022, doi:10.3390/cancers14020324_

Round 1

Reviewer 1 Report

I read with interest the manuscript entitled "Efficacy and Safety of Trans-Arterial Yttrium-90 Radioembolization in Patients with Unresectable Liver-Dominant Metastatic or Primary Hepatic Soft Tissue Sarcomas".

The major aim of the study is of adequate clinical importance and of interest to the audience. Furthermore, the methodology is well-designed and presentation of the work is acceptable. However, my major concern is the following:

  1. The number of the patients included in this study (35) is extremely limited for the conclusions drawn. The conclusions must be extensively revised to present a more cautious interpretation of the safety and efficacy data obtained from a limited number of patients. 

2. The authors should explain why they have excluded the history of systemic chemotherapy, liver surgery or external beam radiotherapy prior to or after TARE have been excluded from the inclusion criteria as this may have a significant impact on the response based on the RECIST criteria. 

Some minor points:

  1. The language needs to be polished since I encountered some ambiguous statements; for instance, in the abstract:

We found that those without evidence of tumor growth after the procedure had longer overall survival, as well as those that had a longer progression-free interval post-procedure. 

Does this mean that, in addition to those without evidence of tumor growth, those with longer PFS post-procedure had longer OS? This does not make sense to me

2. Each table should be interpreted as a stand alone part of the manuscript. The extended forms of some abbreviations such as "DC" and "PD" should be added to the legend of table 3 as has been done for other tables. 

Author Response

Dear Reviewer,

Thank you for your thoughtful comments and review of our manuscript.

Please find below a detailed reply to your comments:

  1. The number of the patients included in this study (35) is extremely limited for the conclusions drawn. The conclusions must be extensively revised to present a more cautious interpretation of the safety and efficacy data obtained from a limited number of patients. 
    • We have revised the discussion and conclusion sections and amended our interpretation of the results. We have tried to emphasize that given the retrospective nature of this single-institution study together with the small size of our patient cohort we cannot draw definitive conclusions regarding the safety and efficacy of TARE in patients with unresectable hepatic soft tissue sarcomas. However, we hope that our study will lay the foundation for further larger multi-institutional clinical studies that will be able to more definitely delineate the role of TARE in patients with hepatic STS.
  1. The authors should explain why they have excluded the history of systemic chemotherapy, liver surgery or external beam radiotherapy prior to or after TARE have been excluded from the inclusion criteria as this may have a significant impact on the response based on the RECIST criteria. 

    • Patients that received systemic chemotherapy, liver surgery or external beam radiotherapy before or after TARE were included in the study since, in current clinical practice, TARE is not employed as a first line treatment strategy for patients with hepatic STS. As a consequence, most of the patients undergoing TARE have already received different cancer-directed therapies before the procedure. Also, many patients receive other cancer-directed therapies after TARE either because of disease progression or to control extrahepatic disease. We have also made a note of this in paragraph 1. of the manuscript. Also, we performed a new analysis of tumor radiologic response measured by both RECIST and mRECIST criteria comparing patients that received other cancer-directed therapies before and after TARE to those that did not receive other therapies. These results are illustrated in paragraph 3.3. of the manuscript and in the supplemental Tables 1 and 2.

  2. The language needs to be polished since I encountered some ambiguous statements; for instance, in the abstract:

We found that those without evidence of tumor growth after the procedure had longer overall survival, as well as those that had a longer progression-free interval post-procedure. Does this mean that, in addition to those without evidence of tumor growth, those with longer PFS post-procedure had longer OS? This does not make sense to me

    • We have corrected the language to the best of our ability across the manuscript. To answer your question above, we found that disease control with no evidence of tumor growth at 6 months post-TARE correlated with longer overall survival, as well as achieving a liver disease progression-free interval equal to or greater than 9 months after TARE. We have changed the simple summary to make this clearer.
  1. Each table should be interpreted as a stand alone part of the manuscript. The extended forms of some abbreviations such as "DC" and "PD" should be added to the legend of table 3 as has been done for other tables. 

    • We have modified the tables legends as requested.

Sincerely,
The Authors

Reviewer 2 Report

The presented paper evaluates the effect of trans-arterial therapies(mainly TARE) on a variety of sarcomas in the liver. The topic is well presented and has a certain clinical relevance. The main drawbacks are already listed by the authors themselfs in the limitation section. Eventhough, the statistic analyses are suffisticated but a little to extensive for the size of the study population. The statistic results therefore mainly have exploratory character. However, the authors present their results accordingly with a clear main statement. For more clarity a differentiation in testing for study hypothesis in contrast to secondary statistics or exploratory statics might be worthwhile.

Minor comments:

1. Please add detail for the multivariate analysis. For what variables the results are correted for?

2. In the discussions section the authors present one case with a 10 year follow-up. This of course rises the question about the age of the date in general. Please consider to omit the notification about 10 years of FU or provide the time interval when the data was gathered.

Author Response

Dear Reviewer,

Thank you for your thoughtful comments and review of our manuscript.

Please find below a detailed reply to your comments:

  1. Please add detail for the multivariate analysis. For what variables the results are corrected for?
    • Only the variables that showed a significant association with increased survival in both the Kaplan Meier analysis and the univariate Cox proportional hazards analysis were included in the multivariate Cox proportional hazards analysis. These included: receiving TACE after TARE; achieving a liver progression-free survival equal to or greater than 9 months, and a Child-Turcotte-Pugh class of B at baseline.
  1. In the discussions section the authors present one case with a 10 year follow-up. This of course rises the question about the age of the date in general. Please consider to omit the notification about 10 years of FU or provide the time interval when the data was gathered.

    • We opted to omit the notification about the 10 years of follow-up.

Sincerely,
The Authors

Round 2

Reviewer 1 Report

The authors have adequately addressed my comments.